# Specificity of Key Sex Determination Genes in a Mammal with Ovotestes: The European Mole *Talpa europaea*

**DOI:** 10.3390/ani14152180

**Published:** 2024-07-26

**Authors:** Alexey Bogdanov, Maria Sokolova, Irina Bakloushinskaya

**Affiliations:** 1Koltzov Institute of Developmental Biology, Russian Academy of Sciences, 119334 Moscow, Russia; bogdalst@yahoo.com (A.B.); mmariasokol@gmail.com (M.S.); 2Biological Department, Lomonosov State University, 119234 Moscow, Russia

**Keywords:** sex determination, ovotestis, European mole, *Sry*, *Rspo1*, *Eif2s3x*, and *Eif2s3y* genes

## Abstract

**Simple Summary:**

Simple Summary: The genetics of sex determination in mammals has been studied mainly in humans and mice. However, some animals develop in other ways, like moles, whose females have mixed gonads that combine two parts: male (testes) and female (ovaries). In other animals and humans, such gonads are known but they are formed due to pathologies and disturb the ability to reproduce up to sterility. We first characterized the structural features of some genes and hypothesized that changes in the structure of the key gene for testis development *Sry* may be responsible for the ovotestes formation in European moles. Here, we considered the presence of the intron-containing fragments of the retroposon and presumably the existing copies of the *Sry* gene along with features of the placenta structure.

**Abstract:**

Here, for the first time, the structure of genes involved in sex determination in mammals (full *Sry* and partial *Rspo1*, *Eif2s3x*, and *Eif2s3y*) was analyzed for the European mole *Talpa europaea* with ovotestes in females. We confirmed male-specificity for *Eif2s3y* and *Sry*. Five exons were revealed for *Rspo1* and the deep similarity with the structure of this gene in *T. occidentalis* was proved. The most intriguing result was obtained for the *Sry* gene, which, in placental mammals, initiates male development. We described two exons for this canonically single-exon gene: the first (initial) exon is only 15 bp while the second exon includes 450 bp. The exons are divided by an extended intron of about 1894 bp, including the fragment of the LINE retroposon. Moreover, in chromatogram fragments, which correspond to intron and DNA areas, flanking both exons, we revealed double peaks, similar to heterozygous nucleotide sites of autosomal genes. This may indicate the existence of two or more copies of the *Sry* gene. Proof of copies requires an additional in-depth study. We hypothesize that unusual structure and possible supernumerary copies of *Sry* may be involved in ovotestes formation.

## 1. Introduction

Sex determination is a fundamental biological process. The presence of two sexes, male and female, with specific types of gonads that produce two types of gametes, provides an opportunity for the exchange of genetic information; sexual reproduction maintains genetic links between generations and ensures the genetic integrity of a species. The formation of gonads as the first and essential event of sexual differentiation is triggered by the interaction of antagonistic genes in the early developmental pathways [1]. The precise timing, location, and regulation of gene expression are necessary for this process [2,3].

In the early 1990s, the crucial role of a member of the SOX family, the *Sry* gene (sex-determining region Y) as a trigger of sex determination in placental mammals, was shown [4,5]. This gene consists of a highly conserved HMG box (high-mobility group DNA-binding domain) and a species-specific and rather variable region outside the HMG box. In mammals, activation of the *Sry* gene located in the Y chromosome initiates the development of testes by upregulation of autosomal *Sox9* [6]. In the absence of *Sry*, such as in the case seen in XX individuals, bipotential gonads differentiate into ovaries probably by initiation of the Wnt4 pathway, which is under positive feedback with *Rspo1* [7,8]. Normal development of the female gonads requires proper regulation of a number of genes such as *Sox9*, *Wnt4*, *Fgf9*, *Rspo1*, and others. These genes, as far as it is known, are localized in the X chromosome or autosomes, i.e., those chromosomes that are also present in males, so the leading role of such factors as a trigger remains problematic. Still, in mammalian species, including humans, some deviations in the gene structure can lead to pathology. If we analyze disorders of gonad development in females with XX, it turns out that few gene mutations lead to the formation of ovotestes, i.e., gonads consisting of tissues characteristic of both ovaries and testes [9]. In the case of the mutations of *Rspo1* (*R-Spondin 1*) gene, ovotestis development was observed in humans with the XX karyotype [10]. In the case of the XY karyotype in mice (the model B6. XY TIR), the absence of *Rspo1* repression can lead to sex reversal [11].

Although the key data on the genetics of sex determination were obtained for humans and laboratory mice, there are other natural or pathological cases of unusual gonad development or sex differentiation gene system functioning. The deviations, existing both as a norm and as a pathology, have received poor attention. Two mutually exclusive cases have been described in different groups of mammals, with one involving the presence of normal sex chromosomes along with the formation of mixed gonads (ovotestes) in females, and the other entailing the absence of *Sry* along with the formation of normal testes in males and ovaries in females. Several species of the *Ellobius* genus (with X0/X0 or XX/XX sex chromosomes) completely lost the Y chromosome and *Sry* [12,13,14]; at the same time, for *Ellobius fuscocapillus*, the only species with standard sex chromosomes in the genus (XX/XY), the presence of the *Sry* gene was described in females [15]. Some species of the *Tokudaia* genus (X0) have partial loss of the Y chromosome and no *Sry* alongside normal testis formation and function in males [16,17].

Alongside these extraordinary rodents, some other mammals obtain unusual gonad structures. Cases of ovotestes presence were reported in freemartin syndrome for bovines in the 18th century [18] and were first explained by Lillie [19] through the hormonal influence of male fetus to female one in monochorionic twins due to vascular anastomosis in early development. Distinct cases of ovotestes formation were described in phenotypically female individuals with the XX genotype (in *Delphinus delphis* [20], dogs, and cats [21,22]) and in phenotypically male individuals with the XY genotype and the presence of the *Sry* gene (in domestic cat [23]). In contrast to the listed cases of pathological gonad development, there are naturally evolved exceptions [24,25]. At least eight mole species of the Talpidae family (XX/XY) possess normal testes in males and ovotestes in females: *Talpa occidentalis*, *T. romana*, *T. europaea*, *T. stankovici*, *Galemys pyrenaicus*, *Condylura cristata*, *Neurotrichus gibbsii*, and *Mogera wogura* [26,27,28,29,30]. To date, it is considered that female gametogenesis occurs in the ovarian part of ovotestis (despite the late formation of the follicular structure and postnatal meiosis), while its testicular part includes androgen-producing Leydig cells, typical for male gonads [31,32,33]. In this case, two sexes are still required for sexual reproduction and no background for hermaphroditism occurs.

We hypothesized that a specific structure of some genes might be a reason for the change in the timeline and chain of genetic events in the embryogenesis of the European mole *Talpa europaea* that led to turning the ovary into ovotestis. We aimed to confirm the genetic structure for two main players—*Sry* and *Rspo1*—and compare specificity with species with typical gonads. For the control of sex and species specificity, we intend to analyze two orthologous genes *Eif2s3* (eukaryotic translation initiation factor 2, subunit 3, structural gene) from X and Y chromosomes, namely *Eif2s3x* and *Eif2s3y,* respectively.

## 2. Materials and Methods

### 2.1. Samples

In total, eight European moles were collected in the Ivanovo and Moscow regions of Russia in 2016–2023. Information about the studied animals and the collection localities is presented in Appendix A. Animals were treated according to conventional international protocols according to the Guidelines for Humane Endpoints for Animals Used in Biomedical Research. All the experimental protocols were approved by the Ethics Committee for Animal Research of the Koltzov Institute of Developmental Biology RAS in accordance with the Regulations for Laboratory Practice in the Russian Federation, the most recent protocol being № 37-25.06.2020. Every possible care was taken to reduce the animal’s suffering during capturing and sampling. Tissue samples and chromosome suspensions were deposited to the Large-Scale Research Facility “Collection of wildlife tissues for genetic research” IDB RAS, state registration number 3579666.

### 2.2. Karyotyping and Molecular Genetic Analysis

To check the reliability of sex determination in moles, we additionally conducted the chromosome analysis in two specimens from our sample (T23-16 and T23-17, see Appendix A). Chromosomes were prepared from bone marrow according to the Ford and Hamerton method [34].

For all specimens, total DNA was extracted from ground heart samples, which were stored in alcohol. A DNA extraction was carried out after treatment with proteinase K; double phenol-chloroform deproteinization with intermediate incubation with ribonuclease A occurred after the first deproteinization phase; and the final precipitation in isopropanol [35].

For our own primer design, for determination of the exon–intron structure, and for interspecific comparisons of the studied genes, we used whole genome sequences from GenBank as well as predicted transcripts and protein-coding DNA fragments for several species of the Talpidae, Soricidae, and Erinaceidae families; sequences of the house mouse *Eif2s3x* and *Eif2s3y* genes were used too (Appendix A). Primers for amplification and sequencing of the entire *Sry* gene (including flanking areas and the internal intron); three non-overlapping fragments of the *Rspo1* gene, which contain all exons along with short introns and flanking areas; and one fragment of the *Eif2s3x* and *Eif2s3y* genes, are listed in Table 1. A polymerase chain reaction (PCR) was carried out in a mixture containing 35 ng of DNA, 2 μL of 10× Taq buffer, 1.6 μL of 2.5 mM dNTP (Sileks, Moscow, Russia), 4 pM of each primer, 1 unit of Taq polymerase (Syntol, Moscow, Russia), and deionized water to a final volume of 20 μL. Amplification was conducted in a TERTSIK thermal cycler (DNA-Technology, Moscow, Russia). PCR included preheating at 94 °C (3 min) and then 35 cycles as follows: 30 s at 94 °C; 1 min at 60–67 °C, depending on the used primer combination (see the exact annealing temperatures in Table 2); 1 min at 72 °C; finally, an extension of the PCR products was performed at 72 °C (6 min). Sanger sequencing was carried out using the ABI PRISM^®^ BigDye^TM^ Terminator v. 3.1 Kit (Applied Biosystems, Foster City, CA, USA) in the AB 3500 genetic analyzer (Applied Biosystems, Foster City, CA, USA) at the Core Centrum of Koltzov Institute of Developmental Biology, Russian Academy of Sciences.

Sequence alignments were performed in Mega X [36]. After alignment, the lengths of three non-overlapping *Rspo1* gene fragments analyzed in all specimens were equal to 251 bp, 485 bp, and ≈1885 bp (an approximate length of the last fragment is due to the difficulty of precisely establishing the number of multiply repeating the same nucleotide in one region, see Appendix A); the fragments of the *Eif2s3x* and *Eif2s3y* genes were 247 bp and 231 bp, respectively (Appendix A); and the total *Sry* gene together with adjacent DNA fragments comprised ≈3584 bp (its approximate length is due to the difficulty of precisely establishing the number of multiply repeating the same nucleotide in two regions, see Appendix A).

The RepeatMasker web tool (https://www.repeatmasker.org/cgi-bin/WEBRepeatMasker, accessed on 9 March 2024) was applied for searching retroposons in obtained sequences. Additional analysis for gene structure was made using the UCSC browser [37]. Sequences of all the genes that we have analyzed have been deposited in GenBank; their accession numbers are presented in Appendix A.

## 3. Results

### 3.1. Karyotyping Results

In karyotypes of both studied specimens of the European mole, we revealed 34 chromosomes, including sex chromosomes. The female T23-17 had two middle-sized metacentric X while the male T23-16 had one X and one dot-like Y (Appendix A). Thus, the karyotype structure in these specimens was very similar to that described previously [38].

### 3.2. The Rspo1 Gene Analysis

Alignment of the predicted mRNA and protein-coding part of the Iberian mole (*T. occidentalis*) *Rspo1* gene with a corresponding whole genome fragment of the same species demonstrate the presence of five exons, which are divided by introns with lengths from 271 bp to more than 10,700 bp (Figure 1a and Appendix A).

A comparison of sequences obtained by us for European moles demonstrated their high similarity, excluding one heterozygotic site in one intron of two specimens (Appendix A). High correspondence of both exons and introns is obvious for the genus *Talpa* species, *T. europaea,* and *T. occidentalis* (Appendix A). More significant differences are observed when comparing genera and families, although the presence of five exons and their homology is quite obvious, for the exclusion of *Suncus etruscus* (Appendix A). Therefore, we note the maintenance of a general structure of the *Rspo1* gene in species, which have ovotestis (the species of the *Talpa* genus and *Condylura cristata*) and do not have it (the species of *Sorex* genus); at the same time, significant differences are revealed between species without ovotestis (*Suncus etruscus* and the *Sorex* genus). These results indicate that differences in the *Rspo1* gene can hardly determine the ovotestis development in moles.

### 3.3. Analysis of the Eif2s3x and Eif2s3y Genes

The alignment of two predicted different mRNA types, which were preliminarily identified as “EIF2S3” and “EIF2S3, X-linked-like”, with the whole genome fragments of the Iberian mole revealed the most similarity of the mRNA with areas from two different linkage groups. Both alignments demonstrate the presence of 12 exons, which are divided by introns with lengths from 83 bp to more than 11,000 bp (Figure 1b,c). The exons from these two different gene types exhibit a high homology with some nucleotide substitutions, which, along with the house mouse *Eif2s3x* and *Eif2s3y* genes, were used by us for the design of primers allowing us to specifically amplify the fragments of these homologous sequences in moles. Amplified fragments comprised parts of the third and fourth exons and the total intron between them. As a result, the DNA fragment, corresponding to the “EIF2S3” mRNA type, was successfully amplified and sequenced in all European moles while the DNA fragment, corresponding to the “EIF2S3, X-linked-like” mRNA type, was successful in males only (Appendix A). Thus, the latter should really be identified as a fragment of the *Eif2s3y* gene localized in the Y chromosome in mammals. This assumption is also supported by the fact that the DNA fragment, corresponding to the “EIF2S3, X-linked-like” mRNA type, is revealed in the same whole genome contig (RCFO01000018) where the *Sry* gene is (see Appendix A and the text below).

Primers designed by us for the *Eif2s3y* gene fragment amplification may be used for the determination of sex in moles, from which only limited material (skin, fixed tissues, cells, carcasses, etc.) is available or whose gonads cannot be reliably identified.

### 3.4. The Sry Gene Analysis

Alignment of predicted mRNA and the protein-coding part of the Iberian mole *Sry* gene with a corresponding whole genome fragment of the same species demonstrated the presence of two exons. The first (initial) exon is very short and consists of only 15 nucleotides while the second exon includes 450 nucleotides. The exons are divided by an extended intron of 1894 bp (Figure 1d).

As in the case of the *Eif2s3y* gene, the *Sry* gene was successfully amplified and sequenced by us, only in males of the European mole (Appendix A); the analysis of this entire gene in *T. europaea* was first conducted. In general, the *Sry* gene appeared to be similar in European and Iberian moles both in exon and intron parts. However, when analyzing chromatogram parts, which correspond to the intron and DNA areas flanking both exons, we unexpectedly revealed double peaks similar to heterozygous sites of autosomal gene sequences (Figure 2). These double peaks are consistently reproduced in chromatograms regardless of the used primers and their combinations. Two of such double peaks were different in two studied *T. europaea* males despite their origin from the same locality; additionally, one more transition (G/T) was revealed inside the intron in these specimens (Appendix A). There were no such double peaks within exons.

A check of the *Sry* gene sequences of European and Iberian moles by means of the RepeatMasker tool revealed fragments of the LINE-L1 retroposon on both sides of the initial exon (Appendix A). In the Iberian mole model, fragments of LINE-L1 and LTR-ERV1 retroposons were also determined in more distant DNA areas, flanking both exons (Appendix A).

A comparison of previously published sequences [39], which correspond to short fragment (158 bp) of the *Sry* gene second exon, demonstrated their complete identity in several species of moles (Appendix A). However, it is surprising that this fragment appeared to be the same also in species that belong to Talpidae and Soricidae families; only the North African hedgehog, *Erinaceus algirus*, has a highly different *Sry* variant. These unexpected results seem to need additional studies of the *Sry* gene in representatives of the Soricidae family.

## 4. Discussion

Through the analysis of the structure of genes presumably involved in gonad formation, we have shown that in *Talpa europaea* characteristics of *Rspo1* and studied fragments of two orthologous genes of X and Y chromosomes, *Eif2s3x* and *Eif2s3y* genes, are rather consistent with other Talpidae and furthermore with the majority of other mammals.

The most intriguing result was obtained for the *Sry* gene. We described the two-exon structure for this canonical single-exon gene (Figure 1d). The exons are divided by an extended intron; fragments of LINE-L1 retroposon are detected in both sides of the initial exon. Moreover, LINE и LTR retroposons were detected in more distant DNA areas, flanking both exons in *T. occidentalis*. The presence of numerous consistently reproduced double peaks, which were located in chromatograms in the intron part of the gene and in adjacent DNA areas to both exons, is difficult to explain because the *Sry* gene is linked with the Y chromosome, which occurs in each cell in a single copy. The phenomenon needs additional analysis, which should be based on both studying European moles from a number of localities and applying special molecular genetic approaches (DNA cloning etc.). Now, we can propose a preliminary hypothesis that the unusual chromatogram pattern might be indicative of the existence of several copies of the *Sry* similar to *Arvicanthis nairobae*, *Lemniscomys barbarus*, *L. massaicus*, *L. rosalia*, *Pelomys fallax*, and *Rhabdomys pumilio* [40]. Since we have only been able to amplify this gene in *T. europaea* males so far and because sequencing of the most extended PCR product did not show any fragmentation, we tend to propose that these presumable *Sry* gene copies are complete and located in the Y chromosome. It is not excluded that their origin might be due to found retroposons. Several copies of *Sry* are usually a set of pseudogenes with a single functioning copy of the gene in some rodents [41,42,43] but six copies of *Sry* in *Rattus norvegicus* are not pseudogenes and at least a few of them are expressed [44].

Recent studies demonstrate that in mice, the *Sry* gene possesses two exons as well [45]. Presumably, the second exon is responsible for stabilizing the protein product and preventing its early degradation. The *Sry* two-exon structure in moles is completely different due to dissimilar segmentation and the specificity of the second exon in mice. In fact, an additional gene fragment with retrotransposon-derived sequences was described as a second exon in mice. In moles, the part of the gene, which was previously considered as a single exon and corresponds to the canonical exon in mice, was split due to the inclusion of an intron saturated with a degenerate mobile element (Appendix A) as in *Microtus cabrerae* [46].

Whereas the structure of *Sry* has not previously been studied in detail, the time of the expression in embryonic development and in adult gonads has been demonstrated [33]; several other genes involved in gonad development and their expression patterns were investigated to identify the cause of ovotestes formation in *T. occidentalis*. The most interesting finding is that *Sox9* and *Amh* expression is completely absent in ovotestes, both in early development and after birth [30,33,47,48]. This does not correspond to the view of normal development of testes in other mammals since it is believed that Sertoli cells (normally expressing *Sox9* and *Amh*) are the first to start differentiation and their formation is critical for maintenance of male-type differentiation of the gonad. Given that in moles the testicular part in the ovotestes grows during the postnatal period and reaches its maximum volume in adult females rather than in early development, it can be concluded that the male pathway is also impaired. In *T. occidentalis*, in both male and female gonads, Leydig cells begin differentiation with the cessation of *Wnt4* expression, suggesting a possible inhibitory role of Wnt4 in Leydig cell differentiation. In normal ovarian development, *Wnt4* is also responsible for inhibiting the development of a testis-like vascular system; however, in *T. occidentalis*, the vascular system of the gonad develops in a testis-like manner in the presence of *Wnt4*. These findings are also supported by studies in mice, in which double deletion of *Wnt4* and *Foxl2* shows masculinization of the sexual system and even the onset of spermatogenesis in the absence of Sertoli cells.

Thus, the causes and mechanisms of the development of ovotestes in moles have not been fully clarified.

Based on our data, we hypothesize that the unusual structure and possible presence of supernumerary complete copies of the *Sry* gene may be involved in the formation of ovotestes in co-twins similar to freemartins. It is known that changes in *Sry* expression in the differentiating gonadal ridge cause developmental abnormalities in gonads [49]. We believe that the developmental peculiarity established in the Talpidae lineage may be related to the presence of the *Sry* gene product abundance (presumably because of several functioning copies of this gene) in placental anastomoses, which are characteristic of the early development of moles. The epitheliochorial type of placenta, which is considered to be the most primitive, is known to be typical for *Talpa* [50]. Similarly, the same type of placenta also arose in Cetartiodactyla, in which ovotestes are known as pathology, in different species of cetaceans, ungulates, and pigs [51]. The mechanism of disturbing development in the freemartins in cattle has remained unclear until now. Jost et al. [52] tried to prove the hypothesis of the hormonal nature, using androgen injections, but failed. So far, the prevailing view is that the developmental disorder in females is due to the delivery through anastomoses of some masculinizing factors produced by the testes of the male twin [53].

The developmental differences in the aforementioned animal groups are significant. Firstly, the number of embryos varies (3–8 for moles, 1–2 for whales, and ungulates). Secondly, for European and Iberian moles, it was shown that up to about day 20 (this is the day when *Sry* activity was shown), embryos are in a space actually providing the barrier-free exchange of nutrients and other substances due to the formation of syncytial structures. It has been suggested that intrauterine cell transfer between human twins can lead to chimerism [54]. As one of the hypotheses, we believe that the circulation of cells, extracellular vesicles with mRNA, and proteins between embryos due to placental anastomoses [55,56] may be the plausible reason for the change in the trajectory of gonad formation in co-twins with different sexes during a critical time window when *Sry* triggers the *Sox9* expression. From this point of view, the ovotestes in Talpidae might be considered as a natural phenomenon similar to freemartinism in cattle and other animals.

## 5. Conclusions

Thus, the specific structure and probable existence of several complete functionally active copies of the *Sry* gene in *T. europaea* might be an evolutionary advantage favoring greater expression of the gene that, coupled with a specific placenta structure, can change female gonad development to ovotestes.

## Figures and Tables

**Figure 1 animals-14-02180-f001:**
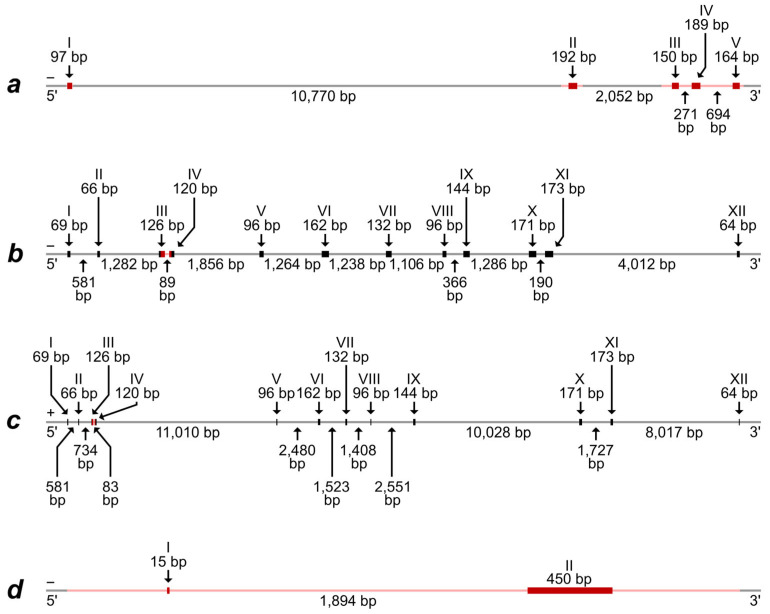
The schemes of exon–intron structure of (**a**)—the *Rspo1* gene; (**b**)—the *Eif2s3x* gene; (**c**)—the *Eif2s3y* gene; and (**d**)—the *Sry* gene. The schemes are based on whole genome sequences, predicted mRNA, and protein-coding DNA areas of *T. occidentalis*. The “positive” and “negative” DNA chains are designated by symbols “+” and “−”, respectively. Exon and intron boundaries are determined approximately because of the same short nucleotide motifs and single nucleotides at the 3′-ends of some exons and adjacent introns. Exons are designated by Roman numerals, with thicker lines and richer shades. Exon and intron lengths are represented above and below the scheme, respectively. DNA areas, corresponding to the sequence in *T. europaea* specimens, are marked by red shades.

**Figure 2 animals-14-02180-f002:**
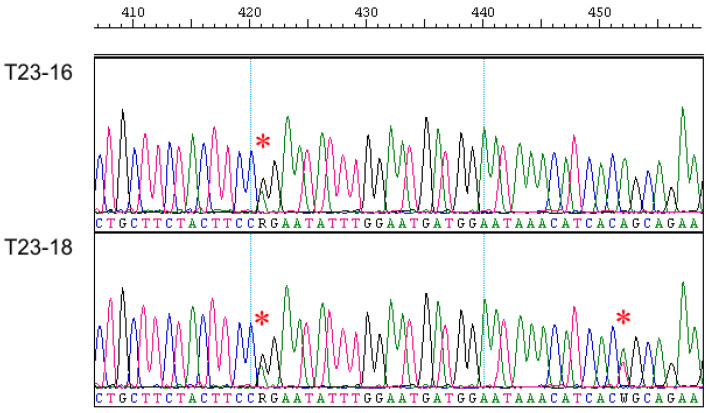
Fragments of the *Sry* gene chromatograms of studied European mole males. Sites with double peaks are marked by an asterisk.

**Table 1 animals-14-02180-t001:** Primers designed by us and used in the study.

Genes and Their Fragments		Primer Sequences, 5′–3′ (Opposite Primers Are Represented in Different Columns)	
*Rspo1*	I fragment (initial)	Rspo1-3T-Fext CAGGCAAGTCGAATCTGCAACGGT	Rspo1-3T-Rext ACTGACCCAGCCCCGGAGGCGTTA
	II fragment (intermediate)	Rspo1-2T-Fext GTGGGTGGGATATCTCAGGTTGTC	Rspo1-2T-Rext GGGGCTCTTGCTGCAGGCGTCTAT
	III fragment (final)	Rspo1-1T-Fext (external) CGTTGACATCCGAGGAATAGAAGT Rspo1-1T-F3ex ATGGTGCCGTTGGCTGCTGTAGAG Rspo1F-Talp CACTGTGCACCTCCGGGTCTCCTT Rspo1-1T-Fint CTGCTGCTGGCCCTTGCGTCTC	Rspo1-1T-Rext (external) GGGTTGCATTGGGTTGGTTTTTGG Rspo1-1T-R3ex CGAGGCCTGCTTCAGCCACAAC Rspo1-1T-R4ex AGCACAATGTGAAATGAGCGA Rspo1-1T-Rint AAGGTGCACGGTGATAAGGACTCC Rspo1-1T-R5ex AGGGCCGGCGGGAGAATGCCAACA
*Eif2s3x*		Eif2s3x-TF GTCATGTAGCTCATGGGAAGTCC	Eif2s3x-TR CTGTAGGAAACTCATCAGGTGTG
*Eif2s3y*		Eif2s3y-TF GGTCATGTTGCTCATGGAAAATCT	Eif2s3y-TR CTGTAGGAAACTCATCAGGTGTA
*Sry*		SRY-TFext (external) AAGTGTAAGTGCAGTGGAAATAAG SRY-TFint2 TACTACAACATGGAAAAACATTAT SRY-Fi2 GACAACGACAGTCCATTACAAG SRY-Fi1 CCTCTTTATGAACACGGACTT SRY-TFint1 AGGTCGATATTTATAGCCCGGGTA SRY-TFint TTAGTTGGCTGTGTTCATGCACT SRY-Fet CAAAACCGTGGCATCATTACAGAA	SRY-TRext (external) TGCCCTTTAAATATCACTAAGGTC SRY-TRst ACCCACTGACTCCAAAACCACAAC SRY-TRint GAATGCATTCATGGTTTGGTCTCG SRY-TRint2 TGAGGTAGCATAAGGGAGAACTGA

**Table 2 animals-14-02180-t002:** Primer combinations, which were used in this study.

Genes and Their Fragments		Primer Combination	Annealing Temperatures
*Rspo1*	I fragment (initial)	Rspo1-3T-Fext/Rspo1-3T-Rext	67 °C
	II fragment (intermediate)	Rspo1-2T-Fext/Rspo1-2T-Rext	67 °C
	III fragment (final)	Rspo1-1T-Fext/Rspo1-1T-Rext	60 °C
		Rspo1-1T-Fext/Rspo1-1T-Rint	60 °C
		Rspo1-1T-Fint/Rspo1-1T-R4ex	60 °C
		Rspo1-1T-Fint/Rspo1-1T-Rext	63 °C
*Eif2s3x*		Eif2s3x-TF/Eif2s3x-TR	63 °C
*Eif2s3y*		Eif2s3y-TF/Eif2s3y-TR	63 °C
*Sry*		SRY-TFext/SRY-TRext	63 °C
		SRY-TFint2/SRY-TRext	60 °C
		SRY-Fi1/SRY-TRext	60 °C
		SRY-TFint/SRY-TRext	63 °C
		SRY-TFint/SRY-TRint	67 °C
		SRY-TFint2/SRY-TRst	60 °C
		SRY-TFext/SRY-TRint	63 °C
		SRY-TFext/SRY-TRint2	63 °C
		SRY-Fet/SRY-TRint2	63 °C

## Data Availability

Sequences of all the genes that we have analyzed have been deposited in GenBank; their accession numbers are presented in Appendix A.

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
