# Peer review of "Specificity of Key Sex Determination Genes in a Mammal with Ovotestes: The European Mole Talpa europaea"

_animals, 2024, doi:10.3390/ani14152180_

Round 1
Reviewer 1 Report
Comments and Suggestions for Authors
This paper examines the genetics of sex determination in gonads of the European mole. The introduction is very well written and clear and shows that the majority of this work has been done before in lab mice. The authors were able to collect samples from 8 European moles. I anticipated that this collection would have been done during pest eradication of moles, but it seems that appropriate ethical standards were employed nonetheless, and this small sample size was able to yield informative results.
The primer design is good and incredibly well detailed, and will be useful in future studies of this species. Although the Rspo1 gene could not determine well the ovotestis development, the primers for reliable sex determination (line 205) will be very useful in further studies.
Overall this paper is incredibly clear and replicable and provides much needed data on species within Eulipotyphla.
Author Response
We are grateful to the Reviewer for taking the time and appreciating our work.
Reviewer 2 Report
Comments and Suggestions for Authors
My comments are in the uploaded PDF file

The manuscript requires a native English speaker to review it from a grammatical and semantic point of view.
Reviewer 3 Report
Comments and Suggestions for Authors
The authors provided very interesting data on the structure of genes involved in sex determination in mammals with ovotestes in females. All the results obtained are highly significant, and I think that the manuscript, after minor revisions, can be considered for publication in the Animals.
Line 85 of Introduction: please check the list of 8 mole species (two same species listed)
Author Response
We are very thankful to the Reviewer for appreciating our work.
It was our mistake Line 85 of Introduction: please check the list of 8 mole species (two same species listed)
We apologize for the mistake. The sentence was corrected with mention of Galemys pyrenaicus instead of one of two T. romana.
Round 2
Reviewer 2 Report
Comments and Suggestions for Authors
The authors have ignored most of my suggestions and unconvincingly refuted most of my criticisms, mainly those related to the subject matter of the manuscript. As a result, the text has not improved sufficiently to justify publication.
Comments on the Quality of English Language
The manuscript requires a native English speaker to review it from a grammatical and semantic point of view.
Author Response
In the response to the first review, we answered all the comments step by step. The lack of clear argumentation in the second review does not allow us to understand what exactly the reviewer did not agree with in our responses. Scientific discussion implies exchange of opinions, but we do not see any feedback in this case.
We would like to accent that the reviewer emphasizes in both the first and second reviews, that our work is phylogenetic one rather than related to the problems of sex determination. Phylogeny, as it is known, is the reconstruction of historical relationships of organisms, populations and taxa of different ranks. Building phylogenetic reconstructions is a special task that requires a special selection of objects, features, and appropriate analysis. Moreover, the databases do not provide sufficient data for phylogenetic constructions on the genes we studied. Phylogenetic study was not planned by us in this short communication.
